# Aptamer Selection Based on Microscale Electrophoretic Filtration Using a Hydrogel-Plugged Capillary Device

**DOI:** 10.3390/molecules27185818

**Published:** 2022-09-08

**Authors:** Junku Takao, Reina Nagai, Tatsuro Endo, Hideaki Hisamoto, Kenji Sueyoshi

**Affiliations:** 1Department of Applied Chemistry, Graduate School of Engineering, Osaka Metropolitan University, Osaka 545-0051, Japan; 2Precursory Research for Embryonic Science and Technology (PRESTO), Japan Science and Technology Agency (JST), Saitama 332-0012, Japan

**Keywords:** aptamer selection, microscale electrophoretic filtration, systematic evolution of ligands by exponential enrichment (SELEX)

## Abstract

This study reports a novel aptamer selection method based on microscale electrophoretic filtration. Aptamers are versatile materials that recognize specific targets and are attractive for their applications in biosensors, diagnosis, and therapy. However, their practical applications remain scarce due to issues with conventional selection methods, such as complicated operations, low-efficiency separation, and expensive apparatus. To overcome these drawbacks, a selection method based on microscale electrophoretic filtration using a capillary partially filled with hydrogel was developed. The electrophoretic filtration of model target proteins (immunoglobulin E (IgE)) using hydrogel, the electrokinetic injection of DNAs to interact with the trapped proteins, the elimination of DNAs with weak interactions, and the selective acquisition of aptamer candidates with strong interactions were successfully demonstrated, revealing the validity of the proposed concept. Two aptamer candidates for IgE were obtained after three selection cycles, and their affinity for the target was confirmed to be less than 1 nM based on their dissociation constant (*K*_D_) values. Therefore, the proposed method allows for the selection of aptamers with simple operations, highly effective separation based on electrophoresis and filtration, and a relatively cheap apparatus with disposable devices.

## 1. Introduction

Aptamers are oligonucleotides that bind strongly to specific target molecules owing to their three-dimensional conformations [1,2,3]. Compared with an antibody that must be biosynthesized, the aptamer possesses many advantages, such as a low cost through artificial synthesis, a low cytotoxicity, and a high applicability to various target molecules, such as large targets (protein [4], virus [5,6,7], and cell [8]), as well as small targets (peptides [9], amino acids [10], and inorganic ions [11]). Thus, aptamers have attracted attention for applications in biosensors [12,13,14], diagnoses [15,16,17], and therapies [18,19,20]. However, for these applications, target-specific aptamers must be experimentally isolated from a large library of oligonucleotides with random sequences.

Aptamer selection from a random library was generally employed in four experimental steps: (i) mixing a target with a library, (ii) separation of the target-oligonucleotide complex from the free/unbound ones, (iii) dissociation of the isolated complexes, and (iv) amplification of the obtained oligonucleotides as aptamer candidates. During the mixing step, the target molecules were added to an oligonucleotide library solution. Oligonucleotides with different sequences showed various affinities for the target molecules under the experimental conditions. Thus, oligonucleotides can form complexes according to the association/dissociation equilibrium. In the separation step, the formed complexes were isolated from the unbound oligonucleotides. In the dissociation step, the addition of denaturants, such as acids, high-concentration salts, surfactants, and organic solvents, weakened the interaction between the target and oligonucleotides, resulting in the dissociation of bound oligonucleotides. After collecting the dissociated oligonucleotides, they were amplified by the polymerase chain reaction (PCR) in the amplification step. The amplified oligonucleotides were used as next-generation libraries containing aptamer candidates. By repeating 10–15 rounds of the selection cycle comprising the above four steps, target-specific aptamers were enriched. This selection strategy, called systematic evolution of ligands by exponential enrichment (SELEX), has become one of the golden standards for aptamer selection [21,22]. The dissociation constants (*K*_D_) of aptamers obtained by SELEX reach the micromolar to picomolar scale, comparable to that of antibodies [23].

In SELEX, a better separation process can reduce the number of SELEX rounds by improving the quality of the amplified libraries, resulting not only in a decrease in the consumption of reagents and time but also in the experimental errors caused by PCR bias, non-specific adsorption, and manual handling [2]. However, insufficient separation leads to contamination by unbound oligonucleotides and greatly degrades the quality of the amplified libraries.

In conventional SELEX, target-immobilized beads are often used for the separation [1,2,3,24]. The target molecule must be immobilized on the bead surface before the beads could be dispersed in the solution containing the oligonucleotide library. After mixing, the oligonucleotides were bound to the immobilized target molecules, forming a complex based on each association/dissociation equilibrium. The complexes on the beads were isolated from the library solution using centrifugal or magnetic forces. Finally, the aptamer candidates were detached from the beads using eluent. Although SELEX techniques using beads have the advantages of simplifying the experimental procedures required to select aptamer candidates, they also require many selection cycles because of the effects of low separation efficiency and non-specific adsorption to the beads. To overcome these limitations, Duan et al. developed a SELEX method using magnetic beads coupled with a DNA library instead of target molecules to suppress the non-specific adsorption of unbound DNAs [25]. The aptamers for histamine and tryptamine were successfully obtained using the proposed method, which requires 17 rounds due to the limited diversity of the library. Wang et al. reported an aptamer-screening device that integrated positive and negative selection units [26]. The developed microfluidic device has two integrated units for packing beads: the upstream or downstream unit is packed with microbeads immobilizing the negative (non-target) proteins or the target protein, respectively. During the introduction of a library solution into the device, only DNAs without binding to non-target proteins at the upstream unit reach the positive-selection beads immobilizing target molecules, which allows successive negative and positive selections in one device. The target-specific aptamer candidates with strong binding could be concentrated at the positive selection unit by introducing only the library solution into the device, and the myoglobin-specific aptamer could be selected only for the seven-round selection.

As described above, SELEX using beads can be achieved through simple procedures, which are still evolving; however, there are numerous limitations. The first is that the binding reaction mainly depends on their diffusion, resulting in a method that requires a long time for binding. Another issue is that they require the immobilization of the target molecules on the bead surface, which leads to structural changes, affecting the binding affinity. Furthermore, non-specific adsorption of oligonucleotides to the beads degrades the quality of the amplified library, which can result in a simple but laborious and costly procedure that consumes large quantities of valuable biological samples.

To overcome these limitations, capillary electrophoresis (CE) has recently been utilized for separation in aptamer selection due its great potential to separate various molecules in a liquid phase with a minimal consumption of reagents [2,27]. Mendonsa et al. first applied CE to the separation step in the SELEX cycles [28,29]. The target molecules were mixed with the DNA library solution and incubated to facilitate binding. Then, the mixture was introduced into the capillary as a short plug, and the complexes were rapidly separated from the free DNAs by electrophoresis. Based on the high separation efficiency in CE, the number of SELEX cycles could be drastically reduced from to 10–15 to 2–4 rounds. Wakui et al. explored a high-affinity and specific aptamer using a separation method combining CE with target-immobilized magnetic beads [30]. The proposed method has the advantages of high recovery using magnetic beads and highly effective separation based on electrophoresis, which demonstrate the selection of aptamers with a high affinity within a few cycles. Saito et al. combined the CE separation of complexes with online sample preconcentration based on transient isotachophoresis [31]. In this method, a large-volume mixture solution containing a DNA library and target microbes are concentrated in the capillary as a narrow zone based on isotachophoresis. Then, the complexes of DNA and microbes are completely resolved from the free DNAs by CE. During electrophoretic migration, the DNAs weakly bound to the target microbes were dissociated, whereas those that were strongly bound remained as complexes. Thus, high-affinity aptamers could be successfully isolated using the proposed method with only one cycle of selection.

This advance in CE-SELEX has markedly reduced the number of rounds of SELEX cycles, although premixing of the target and DNA library in each cycle is required for the equilibrium of association/dissociation. Premixing requires not only a long time owing to the slow molecular diffusion, but also a large-volume consumption of both targets and the DNA library because the optimization of the binding conditions requires many mixtures with different solutions. Zhu et al. studied aptamer selection in the SELEX process without pre-mixing [32]. In the proposed method, plugs of the library and target solutions were individually introduced into the capillary. When a high separation voltage was applied to the capillary, the injected target molecules and DNAs were mixed owing to differences in their electrophoretic mobility. After on-column mixing, the complex and unbound DNAs were rapidly separated by CE. The proposed on-column mixing can easily change the various experimental conditions, which allows the determination of the effects of the aptamer selection environment, such as pH, buffer components, and temperature. In contrast, the number of aptamer candidates obtained decreased with a decrease in the diversity of DNAs that attracted the target molecules in the capillary. Overall, although CE-SELEX is an ideal tool to obtain the desired aptamers, serious problems that require expensive CE equipment and highly skilled technicians remain. Unfortunately, despite their great utility, the number of researchers and technicians able to utilize CE-SELEX is limited, hindering the application of aptamers. Thus, to facilitate aptamer usage as a molecular recognition tool worldwide, a methodology for easier, shorter, more efficient, and less expensive selection is urgently required.

To address this, we propose a novel SELEX method based on microscale electrophoretic filtration (MEF) using a capillary partially plugged with a hydrogel. Our group previously reported on the filtration of enzymes to analyze their kinetics [33]. As previously reported, enzymes incorporated in a hydrogel maintain their enzymatic activity [34,35]; thus, we proposed that the filtrated enzymes also maintain their activity even when they are located inside the hydrogel. The developed technology successfully trapped the target enzymes, alkaline phosphatase (ALP, *M*_W_ = ~140 kDa, p*I* = 4.4~5.8), through the sieving effect of the plugged hydrogel. The activity of the trapped ALP at the interface of the hydrogel could be measured by introducing the substrate molecules by the electrophoretic migration. This revealed that the activity of the trapped ALP did not change significantly compared to that of the ALP in the solution. These results indicated that the conformation of the trapped enzyme was maintained in the proposed selection method, which is based on electrophoretic filtration. Therefore, we applied this technology for aptamer selection (Figure 1). The target molecule was electrokinetically introduced into the capillary partially filled with hydrogel as a trapping step. Based on the molecular sieving effect, target molecules were trapped near the upstream interface of the hydrogel (Figure 1a). After trapping the target molecules, the DNA library was introduced via electrophoresis (the binding step). The DNAs migrated toward the anode and then interacted with the trapped target molecules. According to the association/dissociation equilibrium, DNAs with strong interactions form stable complexes, whereas those with weak or no interactions cannot. Thus, DNAs without strong binding migrated through the hydrogel and were rapidly removed from the capillary (Figure 1b). Only DNAs with binding affinities higher than the threshold were enriched near the interface during the binding step. To obtain surviving DNAs with moderate interactions, the trapped complexes were treated with a washing solution (the washing step). The electrostatic and hydrophobic interactions between the DNAs and target molecules were moderately suppressed by the addition of a salt and surfactant to the washing solution introduced into the capillary. Consequently, the moderately bound DNAs were also dissociated from the complex during the washing step and removed from the boundary (Figure 1c). In the elution step, the eluent containing a higher concentration of salt and surfactant was introduced into the capillary after the washing step. The strongly bound DNAs were dissociated by the eluent before moving toward the downstream reservoir.

In the proposed method, the unbound, weakly interacting, and moderately bound DNAs can be easily eliminated to the anodic reservoir solution during the binding and washing steps, while the strongly bound DNAs remain as a complex near the upstream interface of the hydrogel. In addition, the random DNA library was continuously supplied during the binding step, enriching the amount of bound DNAs to the trapped targets. Moreover, the performance of the obtained aptamers can be easily modulated by adjusting the additive/composition of the washing/eluting solutions. This method will enable aptamer selection with simple experimental procedures and inexpensive systems compared to conventional methods. In this study, a combination of microscale electrophoretic filtration and SELEX (MEF-SELEX) was demonstrated to confirm the proposed methodology.

## 2. Results

### 2.1. Confirmation of Electrophoretic Filtration of Target Molecules

In a previous study, the filtration of the target protein (alkaline phosphatase, ALP; *M_W_* = approximately 140 kDa) was confirmed using a capillary partially plugged with a hydrogel (15 wt% acrylamide (AAm)/bis-acrylamide solution (bis-AAm) [33]. The molecular weight of the model target immunoglobulin G (IgG; *M_W_* = approximately 150 kDa), was larger than that of ALP. Thus, the trapping of IgG using the capillary was tested under weakly basic conditions for the electrophoretic migration of IgG. After the fluorescently labeled IgG (F-IgG) was introduced into the hydrogel-plugged capillary by applying the voltage, the fluorescence intensity near the upstream interface of the hydrogel was significantly increased with increasing time for the electrokinetic injection (Appendix A). This indicates that IgG was successfully filtered and trapped at the cathodic interface of the hydrogel in the capillary using the proposed method. Because the molecular weight of immunoglobulin E (IgE; *M_W_* = approximately 190 kDa) is larger than that of IgG, it is assumed that this method also allows for the trapping of IgE, and an experiment related to aptamer selection was conducted.

### 2.2. Confirmation of the Interaction of DNAs with IgG

After the plugged hydrogel had trapped the IgG, 5′-fluorescein isothiocyanate-labeled DNAs (FITC-DNA) were electrokinetically introduced into the capillary by applying a voltage during the binding step (Figure 2a). Consequently, the fluorescence of FITC-DNAs was observed near the upstream interface of the hydrogel immediately after applying the voltage (Figure 2b,c). The fluorescence intensity profile along the red horizontal dotted line centered in the capillary was estimated using ImageJ software ver. 1.53r (Figure 2d). The fluorescence intensity near the interface increased upon increasing the injection time of FITC-DNA, whereas weak fluorescence in the hydrogel remained during the binding step. This indicated that negatively charged FITC-DNAs migrated toward the anodic side of the capillary, and FITC-DNAs bound to the trapped IgG remained near the upstream interface of the hydrogel, whereas FITC-DNAs without binding passed through the interface. The fluorescence intensity reached a plateau after 10 min of injection with FITC-DNAs (Figure 2d), suggesting that almost the entirety of the IgG could bind with FITC-DNAs. Thus, it was confirmed that DNAs bound to the trapped target were separable from the unbound one using the proposed method.

### 2.3. Evaluation of Effects of Additives on DNA Binding

After the binding step, the anodic reservoir solution was exchanged with one containing the background solution, and a washing voltage was applied for 30 min. Consequently, the fluorescence intensity near the upstream interface of the hydrogel continued to decrease to a steady value (black circle in Figure 3a), indicating the dissociation of some FITC-DNAs and the migration of the released FITC-DNAs toward the anodic side of the capillary. In contrast, many FITC-DNAs still remained near the interface as complexes.

To study the contribution of the additives to electrostatic and hydrophobic interactions during complexation, a background solution containing various concentrations of sodium chloride (NaCl) or sodium dodecyl sulfate (SDS) was used as the washing solution. When NaCl or SDS was added to the washing solution, the decrease in the fluorescence intensity accelerated as the concentration of the additives increased, compared to that without additives (Figure 3a,b). These results indicate that the addition of salt or surfactant suppressed the electrostatic or hydrophobic interactions between the DNAs and target molecules, respectively, under the experimental conditions. Thus, it was clarified that the strongly bound DNAs could survive by adding an appropriate amount of salt and surfactant to the washing solution.

### 2.4. Confirmation of Binding DNAs with IgE

To confirm the effect of the trapped amount of target molecules on DNA binding, IgE and FITC-labeled IgE aptamer [29] (FITC-IgE-Apt) were employed as model samples in the proposed method. After trapping 0.0, 1.0, 1.5, and 2.0 μg/mL IgE for 30 min, the cathodic side of the reservoir solution was exchanged with that containing the background solution, and then 100 V was applied for 5 min to prevent IgE from reaching the hydrogel. FITC-IgE-Apt was then introduced into the IgE-trapped capillaries for 30 min. Thus, a strong fluorescence was also observed near the cathodic-side interface of the hydrogel plug (Figure 4), as described above. After the binding step, the values of fluorescence intensity increased as higher concentrations of IgE were introduced, as shown by the binding step in Figure 4. During the washing step, weakly or moderately adsorbed FITC-IgE-Apt was eliminated by additives. The higher concentration of IgE introduced into the capillary also caused stronger fluorescence to remain after the washing step. This indicates that the higher the introduced concentration of IgE, the greater the amount of trapped IgE and, accordingly, the greater the number of binding sites. When FITC-IgE-Apt was introduced without trapped IgE, on the other hand, the fluorescence increased slightly during the binding step, which indicated that the weak stacking effect was due to the molecular sieving effect of the hydrogel, which decelerated the FITC-IgE-Apt. The observed fluorescence decreased until it was nearly the same as the background fluorescence intensity during the washing step because the FITC-IgE-Apt located near the interface migrated toward the anodic side of the capillary without binding/adsorption. Therefore, it was concluded that the non-specific adsorption of the DNAs to the hydrogel was negligible under the experimental conditions.

When the eluent containing a higher concentration of NaCl than that in the washing solution was introduced into the capillary (elution step in Figure 4), the fluorescence intensities decreased near the background. The disappearance of fluorescence indicated that the strongly bound FITC-IgE-Apt could be eluted from the trapped interface owing to further suppression of the electrostatic interactions. These results confirm that the proposed method allows aptamer candidates to be obtained only by simple electrophoresis, by tuning the concentrations of the additives in the washing solution and eluent.

### 2.5. Selection of IgE Aptamer Candidates by the Proposed Method

Based on the above results and discussion, a 2.0 μg/mL IgE solution prepared with the background solution was employed to increase the amount of bound DNAs. A library solution containing 10 nM random DNAs and 50 mM NaCl was also prepared with the background solution to suppress the binding of DNAs with weak interactions with IgE during the binding step. A background solution containing 100 mM NaCl and 1 mM SDS was prepared as the washing solution to eliminate the DNAs moderately bound to IgE during the washing step. To elute the aptamer candidates effectively, a higher concentration of NaCl (300 mM) was added to the background solution containing 1 mM SDS to suppress both electrostatic and hydrophobic interactions.

To enrich DNAs with strong binding to IgE, the proposed selection process was repeated for three cycles, as shown in Figure 1. After each elution step, PCR was performed to confirm the collection of eluted DNAs. Consequently, the band of the PCR product was observed between the 60 and 80 bp ladders, indicating the successful amplification of the aptamer candidates with a number of their sequence being 78 bp in each PCR product (Figure 5a). This confirmed that the introduction of the eluent allowed for the collection of DNAs that survived the washing step. In each cycle of the proposed method, next-generation sequencing (NGS) using the MiSeq system was carried out after labeling each product (1st, 2nd, and 3rd products in Figure 1) by PCR for sequencing. The obtained sequence data were analyzed using AptamCORE software to explore the aptamer candidates against IgE. As a result, more than two thousand sequences were output by NGS, and the number of major sequences was listed using AptamCORE. In this experiment, the top two sequences, S1 and S2 (Table 1), were specifically enriched during the three-cycle selection, and the ratio of the number of S1 and S2 sequences to that of the total sequence reached over 20% and 10%, respectively (Figure 5b). Repeating the selection also increased the level of S1 and S2, as shown in Figure 5c. Unfortunately, the sequence S2 was not found in the NGS data in the first round because only 5 μL of the eluent (total volume, 500 μL) was used for the NGS analysis in the first round of the experimental procedure. The enrichment factor in each round is defined as follows:

(enrichment factor) = (number S1 or S2 in round *n*)/(number of S1 or S2 in round (*n* − 1)).

The enrichment factor values for the S1 sequence were calculated to be 26- and 6.2-fold in the 2nd and 3rd rounds, respectively. The enrichment value for S2 was calculated to be 7.2-fold in the 3rd round (Figure 5c). As described above, only strongly bound DNAs, owing to their sequence, are able to survive the proposed selection scheme. Thus, DNAs with different sequences were assumed aptamer candidates and their affinity for IgE was verified.

### 2.6. Evaluation of the Sequences and Affinity of the Obtained Aptamer Candidates

The binding affinities of the two aptamer candidates, S1 and S2, were evaluated using CE-LIF to verify the concept of aptamer selection based on MEF. As shown in Figure 6, 5′-FITC-labeled S1 or S2 (FITC-S1, FITC-S2) was detected as a single peak with strong intensity at 100 s. When FITC-S1 or FITC-S2 was mixed with IgE, the signal of each aptamer candidate disappeared, and new peaks were observed at a detection time of 570 s. These results indicate that almost all free FITC-S1 and FITC-S2 were bound to IgE, resulting in the variation of electrophoretic mobility by complex formation. Focusing on the shapes of the newly appeared peaks of the complexes in Figure 6a,b, two clearly separated small signals were observed in the cases of the mixture of FITC-S1 or FITC-S2 and IgE. These results suggest that one or two molecules of S1 or S2 can bind to similar structures in IgE, such as Fab fragments. In addition, the peak areas of these signals were smaller than those of the free FITC-S1 or FITC-S2. The decrease in the total peak area may be due to the interaction of labeled FITC with aromatic amino acids. However, the fundamental reasons for the variations in both the peak shapes and areas were not elucidated in this study, which made it difficult to evaluate the dissociation constants of S1 and S2 under experimental conditions. Experiments using non-equilibrium capillary electrophoresis of equilibrium mixtures (NECEEM) experiments were conducted to evaluate the binding ability of S1 or S2 to IgE. Based on the results of these experiments, the *K*_D_ values of S1 and S2 were estimated to be 0.83 and 1.55 nM, respectively (see Appendix A). This indicated that the selected aptamer candidates, S1 and S2, exhibited strong binding abilities to IgE. Mixtures of FITC-S1 or FITC-S2 and IgG or BSA were also analyzed by CE-LIF to evaluate the specificity of S1 and S2. No significant changes were observed in the electropherograms compared to those obtained by injecting FITC-S1 or FITC-S2 alone (Figure 6). These results indicate that S1 and S2 exhibited specificity to IgE; at least there were no noticeable interactions with IgG or BSA. Consequently, the aptamer candidates selected by the proposed method exhibited both a strong binding ability and specificity to IgE.

## 3. Materials and Methods

### 3.1. Chemical Reagents

IgE purified from human myeloma was purchased from Athens Research & Technology, Inc. (Athens, GA, USA). NaOH was obtained from Kanto Chemical Co., Inc. (Tokyo, Japan). A prepolymer solution containing 30% (*w*/*w*) acrylamide (AAm)/bis-acrylamide solution (bis-AAm) (AAm:bis-AAm = 37.5:1) and IgG from human serum was purchased from Sigma-Aldrich (St. Louis, MO, USA). The reagents included in the polymerase and dNTPs (KAPA HiFi HS ReadyMix) were purchased from Nippon Genetics Co., Ltd. (Tokyo, Japan). Hydrochloric acid, SDS and 2-amino-2-hydroxymethyl-1,3-propanediol were purchased from FUJIFILM Wako Pure Chemical Corporation (Osaka, Japan). The reagents, 2-hydroxy-2-methylpropiophenone (HOMPP) and 3-(trimethoxysilyl)propyl methacrylate, were purchased from Tokyo Chemical Industry (Tokyo, Japan). Xpress Micro Dialyzer MD300 was purchased from Scienove GmbH (Jena, Germany). VB-G25Slide-A-Lyzer dialysis cassettes (2 K MWCO, 0.5 mL) were obtained from Thermo Fisher Scientific Inc. (Waltham, MA, USA). Sodium chloride (NaCl) was obtained from Nacalai Tesque, Inc. (Kyoto, Japan). The HiLyte Fluor^TM^ 555 Labeling Kit-NH_2_ was purchased from Cosmo Bio Co., Ltd. (Tokyo, Japan). The PCR clean-up kit was purchased from Takara Bio, Inc. (Shiga, Japan). A DNA library of random sequences and fluorescently labeled IgE aptamers reported by Shuan et al. (5′-FITC-GACTACCCGGGTATCTAATCCTGAAACATAGCATATTTACTTATGTCGCCTTGCCGGTTCTGCTGCCGCCCGTAGG-3′, *K_D_* = 23 ± 12 nM) was synthesized by Tsukuba Oligo Service Co., Inc. (Ibaraki, Japan). Forward and reverse primers for sequencing and forward and reverse primers for selection were synthesized by BEX Co., Ltd. (Aichi, Japan) and are listed in Appendix A. Deionized water (18 × 10^7^ S/cm) was prepared using a Milli-Q system (direct-Q UV 3; Merck, KGaA, Darmstadt, Germany).

### 3.2. Apparatus

A fluorescence microscope (VB-S20; Keyence Corp., Osaka, Japan) equipped with a light setup (120 W) (VB-L10; Keyence Corp., Osaka, Japan) and a battery (PA250-0.25B; TEXIO Corp., Kanagawa, Japan) was used to observe the fluorescence of the labeled IgE and DNAs in the capillary. A mask aligner (250 W) (UV-CL251S; San-Ei Electric, Osaka, Japan) was used for photopolymerization of the hydrogel. The thermal cycler (TurboCycler) was purchased from Blue-Ray Biotech Corp. (Xindian, Taiwan). A capillary electrophoresis apparatus equipped with a laser-induced fluorescence detector (CE-LIF) (PA800 plus; AB SCIEX LLC, Framingham, MA, USA) was used to evaluate the affinities of the aptamer candidates.

### 3.3. Fabrication of Electrophoresis Device for Aptamer Selection

The preparation of the hydrogel-plugged capillary is described in a previous study [33]. Briefly, a glass capillary (Drummond Scientific Company, Broomall, PA, USA) was cut into 2 cm pieces, and their inner surfaces were activated by sinking them in 0.1 M NaOH solution for 30 min. After washing the capillaries with deionized water and drying them, the inner surface of the capillary was modified by sinking in a solution mixed with 3-(triethoxysilyl)propyl methacrylate and 100 mM HCl (4:1, *v*/*v*) for 3 h. After the modification and washing of the remaining solutions, the modified capillary was dried using a vacuum pump.

To prepare a capillary partially filled with a hydrogel, a 0.2 μL prepolymer solution (50 mM Tris-HCl buffer (pH 9.0) containing 15 wt% AAm/bis-AAm and 1 vol% HOMPP) was introduced into the modified capillary, and the solution was moved to the central part of the capillary by pneumatic pressure using a micropipette. After forming the hydrogel (length: ~2 mm) by UV light irradiation for 2 min, the hollow parts of the capillary were filled with 10 mM Tris-HCl buffer (pH 9.0) as a background solution. In the experiments, a capillary with an inner diameter of 500 or 350 μm was used for the measurements. A thicker capillary was used to observe electrophoretic migration and the filtration of fluorescently labeled proteins and DNA. A narrower one was used for the selection of aptamers to suppress Joule heating during the experiments. As disposable reservoirs, contact holes were punched into the flat top and curved side surfaces of the sampling tubes (650 μL). The fluidic adapter was made of a silicon elastomer (Fukoku Bussan Co., Ltd. Tokyo, Japan) and was attached to the flat top of the disposable reservoirs to suppress solution leakage. The capillary was connected to the reservoirs via connectors, and then each platinum electrode was inserted into the connecting hole located on the curved side surface of the reservoirs. To prevent misalignment during the experiments, the assembled devices were placed on a jig fabricated using a 3D printer (Appendix A).

### 3.4. Confirmation of Electrophoretic Filtration of Target Molecules

To confirm the trapping of target molecules by microscale electrophoretic filtration, IgG (*M_W_*, approximately 150 kDa) was used as a model protein instead of the rare and expensive IgE (*M_W_*, approximately 190 kDa). IgG was fluorescently labeled using a HiLyte Fluor^TM^ 555 labeling kit according to a previous work [33].

After labeling, a fluorescently labeled IgG (F-IgG) solution was prepared with a 10 mM background solution (pH 9.0). The cathodic/anodic reservoirs were filled with a buffer solution with or without F-IgG. F-IgG was then electrokinetically introduced into the hydrogel-plugged capillary by applying a voltage of 100 V for 30 min. The fluorescence near the upstream interface of the hydrogel plug was observed using a fluorescence microscope equipped with excitation/emission filters (RFP for excitation, 540 ± 25 nm; RFP for emission, >572 nm). The fluorescence intensity was calculated from the obtained fluorescence images using the ImageJ software (NIH, Bethesda, MD, USA) (Appendix A).

### 3.5. Confirming the Interaction of DNAs with IgG

To confirm the interaction of DNAs with the target molecules trapped near the hydrogel, IgG and 5′-fluorescein isothiocyanate-labeled random DNAs (FITC-DNAs) were employed as model samples. After the electrokinetic injection of 100 nM IgG at a voltage of 100 V for 30 min, the cathodic reservoir was replaced with that filled with background solution, and a voltage of 100 V was added for 5 min to focus the IgG remaining in the IgG-trapped capillary. During the electrokinetic injection of 10 nM FITC-DNA by applying 100 V for 30 min (binding step), fluorescence near the hydrogel interface was monitored using a fluorescence microscope. After exchanging the anodic and cathodic reservoirs containing the background solution, a washing voltage of 100 V was applied for 30 min (the washing step). During the washing step, fluorescence from F-DNA was observed at the upstream interface of the capillary. Fluorescence imaging during the binding and washing steps was performed using a microscope equipped with filters (GFP-B: 470 ± 40 nm; GFP-B: > 535 ± 50 nm). The fluorescence intensities were also evaluated using ImageJ software.

### 3.6. Evaluation of Effects of Additives on DNA Binding

To assess the effects of the additives in the washing solutions on the interactions between DNAs and IgG, washing solutions containing various concentrations of NaCl (0, 5, 25, 50, and 100 mM) or SDS (0, 0.5, 0.8, and 1 mM) were prepared to control the electrostatic and hydrophobic interactions on complexation, respectively. After the electrokinetic injection of 10 or 20 nM FITC-DNAs into the IgG-trapped capillary, as described above, both reservoirs were exchanged with those filled with the washing solution containing the additives. The fluorescence imaging and estimation of the fluorescence intensity were the same as described in the previous section.

### 3.7. Confirmation of Binding DNA with IgE

The target protein solutions containing IgE were prepared by diluting commercially available IgE solution (1550 μg/mL) with 10 mM Tris-HCl buffer solution (pH 9.0). FITC-labeled IgE aptamer solution (FITC-IgE-Apt, 10 nM) was prepared with 10 mM Tris-HCl buffer solution (pH 9.0). Under the experimental conditions, it was confirmed that the IgE aptamer could specifically bind to IgE by CE-LIF (Appendix A). Each solution with a different concentration of IgE (0.0, 1.0, 1.5, and 2.0 μg/mL) was electrokinetically introduced into each capillary by applying voltages of 100 V for 30 min, resulting in the focusing of IgE onto the upstream interface of each hydrogel. After washing with the background buffer solution for 5 min, the cathodic reservoir was exchanged with that containing FITC-IgE-Apt solution. FITC-IgE-Apt was then introduced into the capillary by applying a voltage of 100 V for 30 min. A washing solution (10 mM Tris HCl (pH 9.0) containing 20 mM NaCl) was employed in the washing step. After the washing step, FITC-IgE-Apt was eluted using an eluent (10 mM Tris HCl (pH 9.0) containing 50 mM NaCl). Among the above processes, fluorescence by FITC-IgE-Apt was observed and evaluated, as described in the experiment using F-DNAs.

### 3.8. Selection of IgE Aptamer Candidates by the Proposed Method

For the selection of aptamer candidates against IgE, 2.0 μg/mL IgE was prepared with the background solution to increase the amount of bound DNAs. A solution containing DNAs with successive 40 random sequences between the forward and reverse primer regions was selected as the library solution. The reagent (10 nM) was prepared with a 10 mM background solution containing 50 mM NaCl without any fluorescent labeling. In the washing step, anodic and cathodic reservoirs filled with 10 mM Tris-HCl buffer solution (pH 9.0) containing 100 mM NaCl and 1 mM SDS were employed to eliminate unbound or weakly bound DNAs by reducing both electrostatic and hydrophobic interactions, respectively. In the elution step, the anodic and cathodic reservoirs filled with 10 mM Tris-HCl buffer solution (pH 9.0) containing 300 mM NaCl and 1 mM SDS were employed for the elution of the strongly bound DNAs to collect the potential aptamer candidates. The other experimental conditions were identical to those described above.

### 3.9. Amplification of DNAs for IgE Aptamer Candidates and Sequence Analysis

As shown in Figure 1, PCR amplification of the eluted DNAs and purification of the amplified products were performed to enrich the aptamer candidates with strong binding. Briefly, the PCR solution was prepared by mixing the eluent (2.5 μL), 2 × KAPA HiFi HS ReadyMix (12.5 μL), 1 μM solution of reverse primer for selection (5 μL), and 1 μM solution of forward primer for selection (5 μL). PCR tubes containing the mixed solutions were placed on a thermal cycler. After the first heating step at 95 °C for 3 min, PCR amplification was conducted by repeating the following steps: denaturation at 95 °C for 30 s, annealing at 60 °C for 30 s, and extension at 72 °C for 10 s. These steps were repeated for 25 cycles, and unreacted sequences were extended completely at 72 °C for 5 min. After PCR amplification, the solution containing the PCR products was purified using a PCR clean-up kit, resulting in the first generation of purified PCR products (1st product). Then, the 1st products solutions were amplified and purified again to increase the amount of the next-generation library required for the proposed selection protocol, providing a second generation of the library (2nd library) solution. As shown in Figure 1, the 2nd products, 3rd libraries, and 3rd products were also obtained by repeating the described cycle, including selection, amplification, and purification. In each cycle, gel electrophoresis of the PCR products was performed using planar agarose gels (3% (*w*/*v*)) to confirm the appropriate amplification. Additionally, each generation of the products was amplified as per the described PCR protocol by using the forward and reverse primers for sequencing instead of those for the selection, resulting in the 1st, 2nd, and 3rd samples for NGS.

### 3.10. Evaluation of the Sequences and Affinity of the Obtained Aptamer Candidates

After preparation of the 1st, 2nd, and 3rd samples, NGS was performed using the Miseq system (Illumina, San Diego, CA, USA) for common use within the Osaka Metropolitan University. The NGS data were analyzed using AptamCORE software to list the aptamer candidates.

To evaluate the binding affinity of the aptamer candidates targeting IgE, listed in Table 1 (S1 and S2), CE-LIF analyses of the fluorescently labeled aptamer candidates were performed. A fused silica capillary (total length: 30.2 cm, effective separation length: 20.0 cm) was used for the analyses. The solutions of the FITC-labeled IgE aptamer candidates, FITC-S1 and FITC-S2, were prepared and mixed with solutions of IgE, IgG, or BSA in 10 mM Tris-HCl buffer solution (pH 9.0). The final concentration of FITC-S1 and FITC-S2 was 10 nM, whereas those of the proteins were 0 (not added) or 100 nM. Before the analyses, 1 M NaOH solution was introduced at a pressure of 20 psi for 3 min to refresh the inner surface of the capillary, and deionized water was then introduced to remove the NaOH solution at a pressure of 20 psi for 3 min. The capillary was then filled with 10 mM Tris-HCl buffer solution (pH 9.0) by applying a pressure of 20 psi for 1 min to precondition the capillary. After preconditioning, each sample solution was injected into the capillary at a pressure of 0.5 psi for 8 s. Both ends of the capillary were inserted into the anodic and cathodic reservoirs containing the background solution, and a separation voltage of 15 kV was applied to the capillary via the reservoir solutions. The fluorescence of the free candidates or complexes was observed at the detection point, revealing their affinity for IgE.

## 4. Discussion

In the proposed method, only three cycles were required for selection, which was less than that obtained by the conventional method using IgE-immobilized beads [36]. Although the number of cycles was comparable to that using capillary electrophoresis [29], CE-based selection requires a much more expensive apparatus and sophisticated knowledge and expertise, whereas the proposed method can be conducted using inexpensive apparatus and disposable devices with simple experimental procedures. Thus, the proposed method is expected to become an ideal tool for the establishment of aptamers that target the desired molecules for capture using a hydrogel.

## 5. Conclusions

In this study, a novel method for aptamer selection based on microscale electrophoretic filtration was developed. The basic concept of the proposed method was confirmed by using IgG and IgE as model targets. The target molecules were successfully trapped by the sieving effect of the plugged hydrogel, which overcame the issues in SELEX using target-immobilized beads. It was also confirmed that the interaction between DNAs and the trapped targets could be controlled by the electrokinetic injection of the DNA library. Online mixing due to fast electrophoresis allowed tuning of the experimental conditions without labor-intensive premixing, which is generally required for conventional SELEX methods owing to slow molecular diffusion. Additionally, the efficiency of the binding was expected to improve because the non-binding DNAs were immediately removed from the interface, trapping the target molecules by electrophoresis. Furthermore, the amount of DNAs that interact with target molecules could be easily increased by extending the injection time.

In the washing step, weakly and moderately bound DNAs could be easily eliminated by tuning the washing solution containing additives. Thus, only the DNAs that can survive during the washing step can be obtained in the elution step, which enhances the quality of the aptamer selection based on the proposed method in each cycle for selection. After three cycles for selecting the IgE aptamer, two types of DNAs with different sequences, S1 and S2, were selectively enriched as aptamer candidates. Both aptamers were found to be specific for IgE and to strongly interact with it, with *K*_D_ values around 1 nM. These findings establish the validity of the proposed method. Compared to other reports, the proposed method provides not only an effective selection of aptamers but also simple experimental procedures using disposable and cheap devices. However, the inner diameter of the capillary (350 μm) was larger than that of the conventional CE-SELEX (50 μm). Thus, the electrophoretic migration of the molecules in the developed device was slower owing to the lower applicable voltage, requiring a short time for binding, washing, and elution for 30 min. In addition, the higher electric current caused Joule heating and electrolysis, such that large-volume reservoirs (500 μL) had to be used to reduce their effects. Although issues remain and the proposed method is not yet ideal for aptamer selection, these can be overcome easily by integrating the proposed method on a microfluidic chip. Aptamer selection using a microfluidic device integrating MEF will simplify the experimental procedure, reduce the consumption of reagents, and shorten the total time, which will also facilitate the improvement of the specificity of the selected aptamer by further integration to connect the negative selection device, as previously reported.

## Data Availability

Not applicable.

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
