# Peer review of "Aptamer Selection Based on Microscale Electrophoretic Filtration Using a Hydrogel-Plugged Capillary Device"

_molecules, 2022, doi:10.3390/molecules27185818_

Round 1

Reviewer 1 Report

molecules-1801846

The authors demonstrate a method for selecting aptamers based on a microscale electrophoretic filtration system. In this method a hydrogel is deposited in a capillary to “trap” targets of interest and expose them to a pool of DNA sequences to identify target-binding aptamers. The hydrogels is proposed to allow the flow of unbound DNA sequences and provide an effective way for separation of binder and non-binder sequences. The manuscript is, in general, well-written and it follows a logical sequence of developments to demonstrate the ability to select aptamers. There are a few major issues the authors need to address before the manuscript can be recommended for publication, as described below:

1.       One of the methods the authors compared their approach to is the use of immobilization of targets in magnetic particles for ease of separation. The authors make the case that one disadvantage of that method is that the target is typically immobilized on the particles which can result in issues with affinity due to blocking of binding sites, denaturing, etc.. It is important to notice that in the proposed method, presumably, the targets are adsorbed in the hydrogel and therefore, it is to a significant degree, immobilized on a surface, which can potentially result in some of the same issues as with the magnetic beads approach the authors referred to. The authors need to discuss this and/or clarify how this method is a superior alternative/prevent this issues.

2.       The hydrogel is the critical component to provide the retention of the target and allow DNA binding, there isn’t really characterization of the interactions between the target and the polymer/hydrogel here, does it work the same with proteins of different molecular weights, isoelectric points, etc? If the authors are making the case that this platform can be used for different targets all these questions are extremely important and should be addressed experimentally. Additionally, the scheme shows the IgG adsorbed to the external part of the interface of the hydrogel and the media, is there any retention inside the hydrogel?

3.       On page 5 the authors hypothesized that low affinity DNA sequences are eluted during washing steps, how can they tell these are not just adsorbed to the hydrogel, instead of binding the target, if the authors want to make this claim they need to collect some of those sequences and show that they bind weakly to the target

4.       Figure 4 shows data that is worrisome in terms of DNA sticking to the hydrogel matrix, at the 0 ug/mL IgE concentration a significant amount of DNA is deposited in the capillary as shown by F.I. of ~ 40 (a. u), the authors mentioned in their introduction how non-specific interactions are detrimental in aptamer selection experiments, this is concerning.

5.       Figure 5 shows the “ratio of amplified S1 and S2”, to understand better the selection process in this new method, they should report the pool enrichment in each round.

6.       It is not clear that the experiment in Figure 6 is the most appropriate. It is clear that addition of IgE to the selected aptamer results in changes in the retention time of the aptamer, however, a valid question is, were the same conditions applied to the other targets? It is this reviewer understanding that the conditions to run a protein with different MW and isoelectric point in a CE experiment need to be optimized, so, are these experiments really comparable?

7.       Finally, the authors need to perform a proper Kd determination for the selected sequences, there is no way an aptamer selection paper can be published without proper binding affinity data.

Author Response

Regarding to the comment 7, we also think that the determination of the proper Kd values are very important. According your valuable comment, we are trying to estimate their Kd values by some experiments using CE-LIF. Unfortunately, we do not have the CE-LIF apparatus, and are in the process of applying to use the equipment. Please wait for a few weeks to complete the experiments and revise the manuscript.

Reviewer 2 Report

The mansucript reports designing a method for aptamer selection using microscale electrophoretic filtration method (MEF). Hydrogel was used in the end of a capillary electrophoresis system as a traping unit. The target molecules (IgE) were captured by the hydrogel and SELEX library was moved electrophoretically over the trapped target molecules. Affinity binding is used to collext aptamer candidates.  The bound libarary members were eluted and sequenced by NGS method. Two aptamer candi,dates were cahracterized for binding affinity, LOD and affinity constant. The specificity was checked by BSA and IgG. This is an original contribution. In fact, there is a cappliary electrophoresis based SELEX methods reported previously. However, this manuscript makes the application easier by including filtration trap. I would like to point out that the method as descibed in the manuscript does not include any negative or counter selection, which might lead to low specificty aptamer selection. But, the seleted IgE aptamers with MEF-SELEX produced a spesificity over BSA and IgG. The manuscript is genearally well written and the results were presented and discussed properly. 

Author Response

Thank you for your valuable comments. As above commented, the application of negative (counter) selection will make improve specificity/selectivity of the obtained aptamer. The proposed scheme will be applicable to negative-to-positive successive selection by simply connecting the developed device, which is in progress at our laboratory. In the near future, we would like to contribute Molecules by submitting the sophisticated MEF-SELEX using the improved devices.

Round 2

Reviewer 1 Report

Authors have requested extra time to finish experiments

Author Response

The original comment offering an extension of the revision deadline for the experiment was entered on June 29, before the original deadline. For systemic reasons in the web-form, we were unable to make the relevant comment change, but as of the August 11 resubmission, the experiment has been completed and its results have been reflected in the revised manuscript and Supporting Information. We would appreciate it if you could review Section 2.5 to ensure that the content is acceptable.

Regarding your comment that minor English corrections are needed, we will request another proofreading by a native English speaker.

Round 3

Reviewer 1 Report

the authors have addressed the issues mentioned by this reviewer